# Breathing porous liquids based on responsive metal-organic framework particles

Athanasios Koutsianos[1], Roman Pallach [1], Louis Frentzel-Beyme [1], Chinmoy Das [1], Michael Paulus[2], Christian Sternemann [2] & Sebastian Henke [1] ✉

Responsive metal-organic frameworks (MOFs) that display sigmoidal gas sorption isotherms triggered by discrete gas pressure-induced structural transformations are highly promising materials for energy related applications. However, their lack of transportability via continuous flow hinders their application in systems and designs that rely on liquid agents. We herein present examples of responsive liquid systems which exhibit a breathing behaviour and show step-shaped gas sorption isotherms, akin to the distinct oxygen saturation curve of haemoglobin in blood. Dispersions of flexible MOF nanocrystals in a size-excluded silicone oil form stable porous liquids exhibiting gated uptake for $CO_2$, propane and propylene, as characterized by sigmoidal gas sorption isotherms with distinct transition steps. In situ X-ray diffraction studies show that the sigmoidal gas sorption curve is caused by a narrow pore to large pore phase transformation of the flexible MOF nanocrystals, which respond to gas pressure despite being dispersed in silicone oil. Given the established flexible nature and tunability of a range of MOFs, these results herald the advent of breathing porous liquids whose sorption properties can be tuned rationally for a variety of technological applications.

Liquids are typically non-porous (i.e. do not own persistent void space). As such, liquids display a nearly linear increase of gas solubility with increasing gas pressure consistent with Henry's law (Fig. 1a)[1]. Nature, however, provides an impressive example of a liquid with a step-shaped gas saturation curve. Owing to a cooperative structural transformation of haemoglobin upon $O_2$ binding, the $O_2$ saturation curve of blood displays a characteristic sigmoidal shape as a function of the oxygen partial pressure (Fig. 1b)[2]. The sigmoidal shape of the $O_2$ saturation curve of blood is of the highest relevance for an efficient $O_2$ transport mechanism from the respiratory organs to the tissues and thus a key factor for metabolism in almost all vertebrates.

The family of *porous liquids* (PLs) entered the stage a few years ago[3,4]. PLs are synthetic liquid systems containing persistent voids capable of hosting gas molecules via the process of physisorption[4,5]. This is different from a chemical reaction of the liquid with a gas as for example utilized in $CO_2$ scrubbing by amine solutions[6] and $CO_2$ "chemisorption" by ionic liquids (ILs)[7]. In a physisorption process, only weak interactions between the gas and the sorbent drive the sorption process, resulting in better reversibility of the process and a lower energy penalty for recycling. Four different categories of PLs have been realized so far[8]: Neat liquid substances including intrinsic "holes" in their molecular structure are defined as type 1 PLs. Type 2 PLs are two-component systems, comprising molecular hosts with intrinsic porosity dissolved in size-excluded solvents. Dispersions of solid microporous particles in sterically hindered solvents constitute type 3 PLs, while strongly associated network liquids displaying intrinsic

[1]Anorganische Chemie, Fakultät für Chemie und Chemische Biologie, Technische Universität Dortmund, Otto-Hahn-Straße 6, 44227 Dortmund, Germany. [2]Fakultät Physik/DELTA, Technische Universität Dortmund, Maria-Goeppert-Mayer Str. 2, 44221 Dortmund, Germany. ✉e-mail: sebastian.henke@tu-dortmund.de

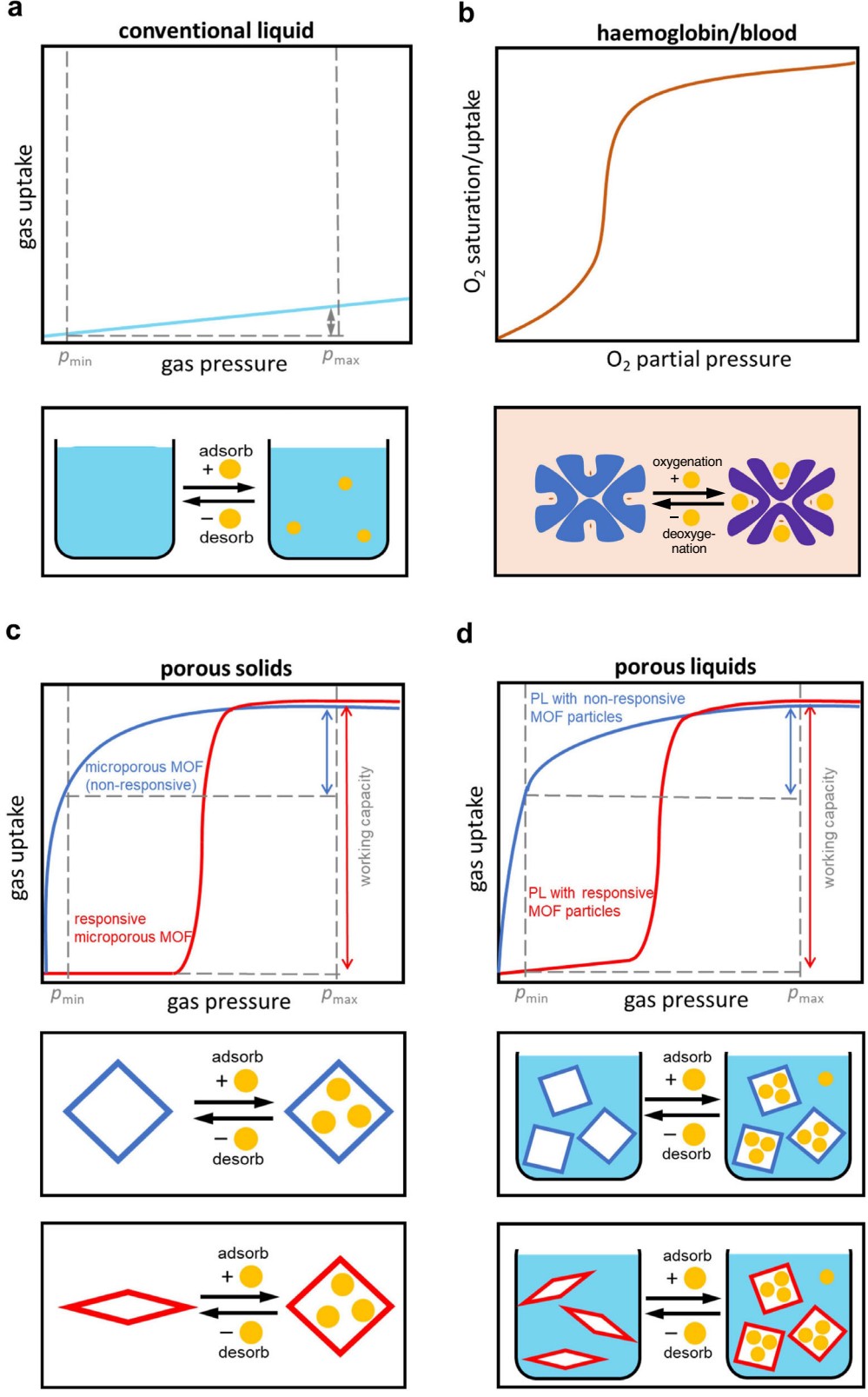

**Fig. 1 | Illustration of the gas sorption behaviour of liquids, porous solids, and porous liquids. a** Isothermal gas solubility curve of typical liquids. **b** Isothermal oxygen binding curve of haemoglobin. **c** Sorption isotherms of non-responsive (blue) and responsive (red) porous solids (i.e. microporous MOFs). **d** Isothermal gas uptake of PLs with non-responsive MOF particles (blue) and responsive MOF particles (red). Maximum adsorption pressure ($p_{max}$) and minimum desorption pressure ($p_{min}$) for a pressure swing are indicated by dashed vertical grey lines.

porosity despite their dynamic network structure represent type 4 PLs[8]. Owing to their unusual combination of permanent porosity and fluidity[9], PLs have emerged as promising materials for gas capture[10,11], molecular separation[12–14], and catalytic applications[15].

With several microporous particles on the market and the wide range of liquid solvents available, type 3 PLs are currently the most economical and scalable[14,16,17]. The sorption capacity of these PLs typically is the weighted average of their constituent phases, i.e. the porous solid particles and the surrounding size-excluded liquid[8,14]. Particularly, metal-organic frameworks (MOFs) have been employed as microporous solid hosts for the development of type 3 PLs[11,13,14,18–21]. MOFs are a versatile class of highly porous, crystalline solids composed of inorganic building units (metal cations or metal cation clusters) which are interlinked to open frameworks by organic building units, the linkers. Their robust nature makes MOFs promising candidates for a variety of technological applications ranging from gas storage, separations, and catalysis to medicine and energy storage[22–24].

Most MOFs feature a microporous network structure resulting in a characteristic gas sorption isotherm of Langmuir shape (Fig. 1c)[25]. Advancements in the field have led to the development of responsive MOFs that undergo structural transitions upon the adsorption of gas molecules, a process often referred to as *breathing*[26–28]. Such gas-pressure-induced breathing phase transformations from a contracted barely porous MOF phase to an expanded MOF phase with significantly higher porosity result in adsorption profiles with a sigmoidal shape (Fig. 1c). Owing to this sorption mechanism, such responsive MOFs exhibit large variations in gas uptake over a very narrow pressure range and hence higher working capacity in a small operating pressure range[29,30].

Research on PLs in recent years almost exclusively focused on the formulation of type 3 PLs containing non-responsive MOF particles[14,31,32]. Even though such formulations led to significant increases in gas uptake capacities in these PLs, studies on the deliberate utilization of responsive MOF particles, aiming for PLs exhibiting intriguing sigmoidal gas sorption curves, are scarce (Fig. 1d). To the best of our knowledge, only two examples of PLs containing responsive MOF particles has been reported[13,33]. Very recently, Brand et al. reported an IL-based PL including photoresponsive MOF particles, which allowed releasing about 8% of adsorbed $CO_2$ molecules by light irradiation[33]. However, the sorption isotherm of the photoresponsive PL exhibited a typical linear shape. Previously, Lai et al. observed a step-shaped ethene sorption isotherm for a sesame-oil-based PL containing ZIF-7 particles (ZIF-7 = Zn(bim)$_2$, bim$^-$ = benzimidazolate, ZIF = zeolitic imidazolate framework)[13]. The stepped shape isotherm was rationally attributed to the gas-pressure-responsive behaviour of the ZIF-7 particles, however, only very few data points have been recorded in the step region of the isotherm. Besides these PLs, amine-containing non-porous organic liquids displaying sigmoidal $CO_2$ sorption isotherms have been developed. These systems, however,

rely on a chemical reaction of the amine group of the organic solvent with $CO_2$, thus requiring high temperatures for the regeneration of the absorbent (i.e. the release of chemically bound $CO_2$) in practical applications[34].

In this work, we investigate the sorption behaviour of breathing PLs based on stable dispersions of particles of the responsive MOFs ZIF-7 and ZIF-9 (Co(bim)$_2$) in size-excluded silicone oil. Via high-resolution gas sorption experiments, also at variable temperatures, we demonstrate the persistence of a breathing behaviour toward $CO_2$ and certain hydrocarbon gases (propane and propylene) for the responsive ZIF particles, even though the particles are dispersed in a silicone oil of a rather low sorption capacity. Importantly, we are able to follow the gas-pressure-triggered phase transition of the ZIF particles in situ by synchrotron radiation X-ray diffraction (XRD), providing the first experimental evidence for the breathing of MOF particles in a liquid dispersion. The responsive PLs exhibit a unique combination of flowability and pumpability with a gas sorption mechanism characterized by a sigmoidal gas saturation curve. Such MOF-based responsive PLs could revolutionize gas separation processes and pave the way for the development of synthetic PLs with high working capacities at sub-ambient pressures, which are particularly relevant for several applications.

## Results

For the preparation of breathing PLs of type 3, the solid MOF constituent must fulfill the following requirements:

(i) The MOF must present a switching behaviour between a non-porous narrow pore (*np*) and a porous large pore (*lp*) phase depending on the applied gas pressure.

(ii) The pore openings of both the *np* and *lp* phases must be sufficiently small to prevent penetration of the host liquid into the pores, so that the intrinsic pores of the MOF particles cannot be occupied by the host liquid, but just by the gas molecules.

(iii) The MOF materials must be accessible as small enough submicro- or nanoparticles, in order to allow the formulation of stable dispersions.

For the proof of concept outlined here, the isostructural MOF materials ZIF-7 and ZIF-9 were selected for the formulation of responsive type 3 PLs[23]. Both frameworks feature the same **sod** (sodalite) topology, and are known to undergo a phase transition from a nearly non-porous *np* phase to a porous *lp* phase (Fig. 2a) upon the adsorption of various gases (for example $CO_2$ and different hydrocarbons), and further fulfill the other requirements for the preparation of responsive PLs given above[35–38]. Moreover, the previous study by Lai et al. already demonstrated that at least ZIF-7 is a suitable material for the development of breathing PLs[13]. The sorption-triggered *np*–*lp* phase transition of these ZIFs leads to sigmoidal sorption isotherms with the specific transition pressure depending on the nature of the gas as well as the type of metal ion in the ZIF. The Co-based ZIF-9

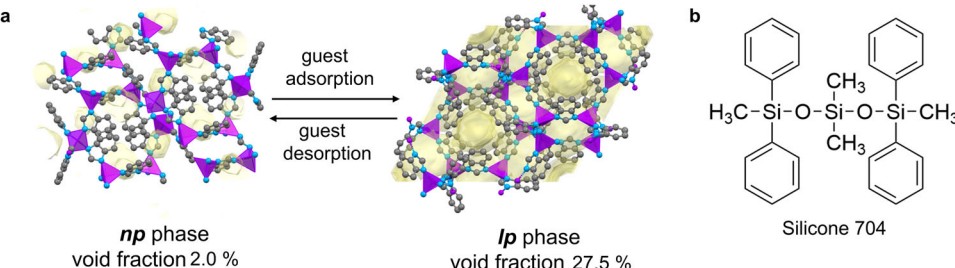

**a**

*np* phase
void fraction 2.0 %

guest adsorption
guest desorption

*lp* phase
void fraction 27.5 %

**b**

H$_3$C–Si–O–Si–O–Si–CH$_3$

Silicone 704

**Fig. 2 | Representation of the structures of the PLs constituents. a** Crystal structures of the *np* (CCDC code RIPNOV01[61]) and *lp* phases (CCDC code VELVIS[23]) of ZIF-7. Purple, blue, and grey spheres depict Zn, N, and C atoms, respectively, while H atoms are omitted for clarity. The voids are shown in pale yellow highlighting the difference in accessible porosity between the *lp* and *np* phases. A probe radius of 1.2 Å and grid spacing of 0.2 Å were applied. **b** Molecular structure of Silicone 704.

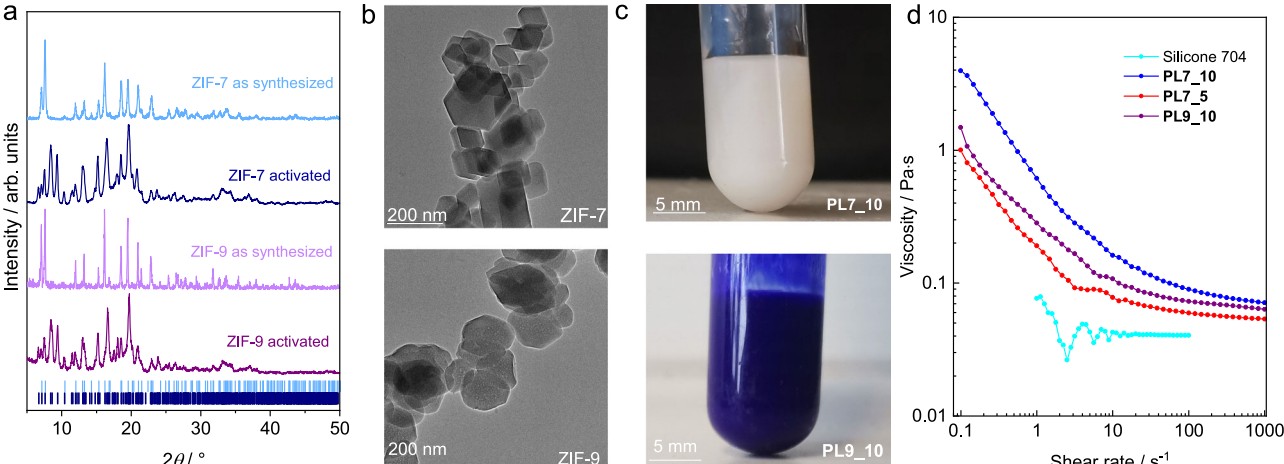

**Fig. 3 | Structural and rheological characterization of the PLs and their constituents. a** PXRD patterns of as synthesized and activated ZIF-7 and ZIF-9 nanocrystals. Tick marks represent the characteristic Bragg reflection positions of ZIF-7 ***np*** phase (dark blue)[61] and ***lp*** phase (light blue)[23]. **b** TEM images of ZIF-7 and ZIF-9 nanocrystal aggregates. **c** Photographs of the dispersions of ZIF-7 and ZIF-9 nanocrystals in Silicone 704, **PL7_10** and **PL9_10**. **d** Graph of the dynamic viscosity vs. shear rate for Silicone 704 and the corresponding ZIF-containing PLs at 25 °C and ambient pressure.

generally exhibits higher phase transition pressures for a particular gas than the Zn-based ZIF-7.

With the aim to utilize ZIF-7 and ZIF-9 as active media in PLs, both materials were synthesized in the form of nanocrystals as described in the "Methods" section in detail[39,40]. Powder (P)XRD data confirmed the formation of phase pure ZIF-7/-9 in the solvated ***lp*** phase (Fig. 3a, profile fits are shown in Supplementary Figs. 3 and 4) and their phase transition to the ***np*** phase upon removal of solvent species (vacuum activation, Fig. 3a and Supplementary Figs. 3 and 4). Crystal size control, and downsizing specifically, are pivotal factors for the development of stable dispersions, with the reduction of crystal size enhancing the stability of MOF-based PLs[14]. In turn, downsizing below critical dimensions can dramatically affect the responsiveness of flexible MOFs and may even lead to the absence of responsive behaviour for very small nanocrystals[41]. According to transmission electron microscopy (TEM) and scanning electron microscopy (SEM) images, the prepared ZIF-7 and ZIF-9 samples consist of primary nanocrystals featuring sizes ranging from 100 to 200 nm (Fig. 3b, Supplementary Figs. 8 and 10), which are integrated into larger aggregates. Nevertheless, the size of the ZIF-7/-9 nanocrystal aggregates was small enough for the preparation of stable PLs, while the crystals were still large enough to exhibit the typical guest-responsive properties as confirmed by the relevant PXRD data (Fig. 3a and Supplementary Figs. 3 and 4).

The commercial silicone oil 1,3,3,5-tetramethyl-1,1,5,5-tetraphenyl-trisiloxane (Silicone 704, Fig. 2b) was used as the liquid phase (host medium) for the preparation of the ZIF-7- and ZIF-9-based responsive PLs. Silicone 704 is a bulky liquid that is size excluded from the pores of the ZIFs and exhibits a particularly low vapour pressure (~$10^{-7}$ Pa at 25 °C) and a rather low dynamic viscosity (about 42 mPa s). The low vapour pressure of the host medium is important for the application in gas adsorption as it minimizes fluid loss under low-pressure conditions. In turn, the low viscosity of the host medium is required to ensure the retention of fluidity after the suspension of the solid ZIF nanocrystals. Dispersions of 5 wt% and 10 wt% ZIF-7 in Silicone 704, hereinafter denoted as **PL7_5** and **PL7_10** were prepared by mixing the corresponding masses of dried ZIF particles and silicone oil followed by sonication for 15 min. Similarly, a dispersion of 10 wt% ZIF-9 in Silicone 704, denoted as **PL9_10**, was prepared (Fig. 3c).

The formulated PLs proved to be stable and did not display any sedimentation, at least after 3 months of standing without agitation (Supplementary Fig. 16). Thermogravimetric analysis (TGA) of the PLs

reveals no significant mass loss up to 200 °C, consistent with the reported low vapour pressure of Silicone 704 and the high thermal stability of ZIF-7 and ZIF-9 (Supplementary Fig. 18). In the range from 200 °C to ca. 325 °C the Silicone 704 is completely evaporated. The residual masses at 350 °C approached the values expected from initial ZIF loading, demonstrating that only the pure ZIF particles are left behind. In accordance with previous reports, the ZIF particles start decomposing from ca. 450 °C[42,43].

The influence of the ZIF particles on the rheological properties of the liquids was investigated by rheological measurements with a rotational rheometer applying shear rates from 0.1 to 1000 s$^{-1}$ at 25 °C (Fig. 3d). The measured dynamic viscosity of Silicone 704 comes in agreement with the values specified from the manufacturer (37–42 mPa s) being equal to 40 mPa s at 25 °C for shear rates higher than 10 s$^{-1}$. Variations in the viscosity of Silicone 704 at shear rates lower than 10 s$^{-1}$ originate from measurement errors due to the limitation of the setup. The Newtonian nature of the silicone oil over the entire range of shear rates is clearly displayed at −10 °C, where the measured viscosity is independent of the shear rate (Supplementary Fig. 12). As expected **PL7_10** and **PL9_10** displayed significantly higher dynamic viscosities compared to the neat silicone oil. At a shear rate of 10 s$^{-1}$, **PL7_10** and **PL9_10** feature viscosities of 161 and 107 mPa s. Typical for suspensions of solid nanoparticles, shear thinning behaviour characterized by decreasing viscosity with increasing shear rates is observed for all PLs, so that viscosities of 71 and 63 mPa s are approached for **PL7_10** and **PL9_10** at a shear rate of 1000 s$^{-1}$. It is anticipated that the agglomerated ZIF nanocrystals tend to separate into primary crystals or smaller agglomerates under shear stress, resulting in reduced flow resistance and the observed shear thinning behaviour[9,44]. The persistence of larger ZIF nanocrystal aggregates in solution is supported by dynamic light scattering (DLS) measurements of **PL7_10**, revealing an average particle diameter of 469 ± 74 nm (Supplementary Fig. 13). Naturally, **PL7_5**, the PL having a lower concentration of ZIF-7 nanoparticles, generally displays lower dynamic viscosities over the entire range of investigated shear rates approaching 54 mPa s at 1000 s$^{-1}$.

Isothermal gas sorption measurements of the derived PLs as well as their constituents (i.e. the pure ZIF nanocrystals and the pure silicone oil) were performed using a conventional volumetric gas sorption apparatus. Modifications of the gas sorption apparatus, originally manufactured for measurements on solid samples, were necessary for the collection of gas sorption isotherms of liquids under continuous

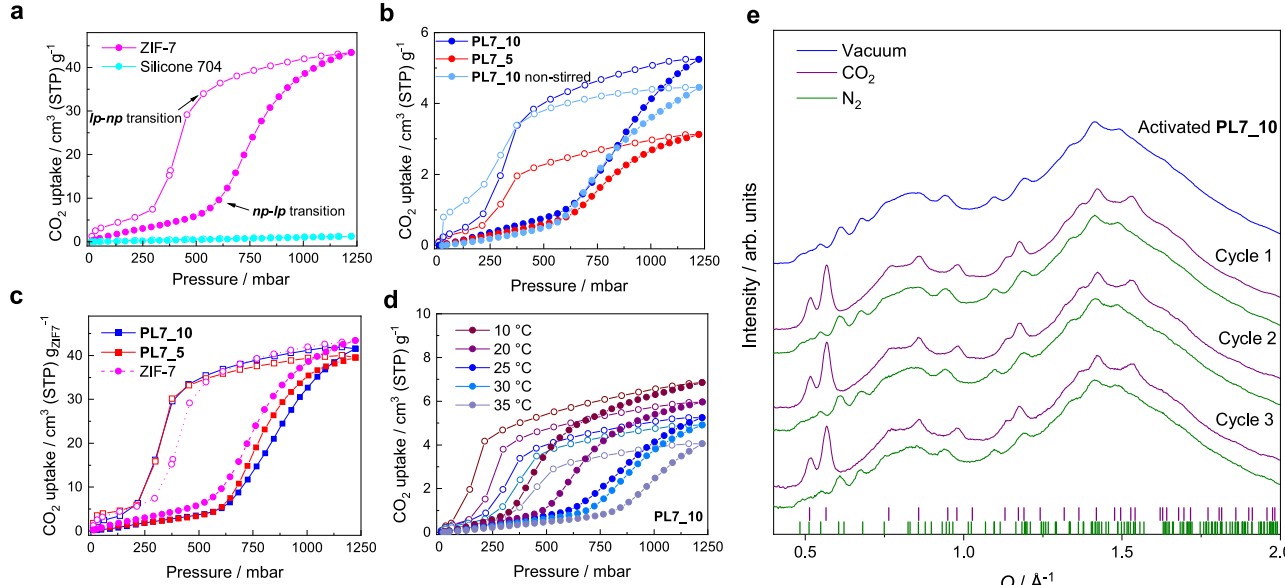

**Fig. 4 | Gas uptake behaviour of PL7 materials. a** $CO_2$ sorption isotherms of ZIF-7 nanoparticles (magenta) and Silicone 704 (cyan) at 25 °C. The regions of the phase transitions of ZIF-7 are highlighted. **b** $CO_2$ sorption isotherms of **PL7_10** (blue circles), **PL7_10** recorded without stirring (light blue circles) and **PL7_5** (red circles) at 25 °C. **c** $CO_2$ sorption isotherms of **PL7_10** (blue squares) and **PL7_5** (red squares) per unit mass of ZIF-7 in comparison to the pure ZIF-7 nanocrystals (magenta circles). **d** $CO_2$ sorption isotherms of **PL7_10** at different temperatures ranging from 10 to 35 °C. For all isotherms, the adsorption branch is shown with filled symbols, and the desorption branch with empty symbols. Lines are just a guide to the eye. **e** Synchrotron radiation XRD patterns of activated **PL7_10** (blue), conditioned under approx. 1 bar $CO_2$ (purple) or $N_2$ (green) for three consecutive cycles. Tick marks represent the characteristic Bragg reflection positions of the $CO_2$-saturated ***lp*** phase of ZIF-7 as extracted by profile fitting (purple; Supplementary Fig. 6 in Supplementary Figures—Powder X-ray diffraction) and the ***np*** phase of ZIF-7 (green; CCDC code RIPNOV01). Only the lower $Q$ region is displayed to highlight the differences. The full dataset of the collected patterns can be found in Supplementary Figures—In situ XRD.

stirring (more information is provided in Supplementary Methods-Gas sorption measurements). In accordance with previous reports[42], the pure ZIF-7 nanocrystals feature the desired sigmoidal $CO_2$ sorption isotherm with a prominent step at about 605 mbar $CO_2$ pressure on the adsorption branch and a capacity of ca. 43 $cm^3$ (STP) $g^{-1}$ at 1224 mbar and 25 °C (Fig. 4a). The step in the adsorption isotherm is the trademark of the ***np–lp*** phase transition of ZIF-7 during $CO_2$ adsorption. The reverse ***lp–np*** phase transition of the materials occurs at a pressure of 456 mbar on the desorption branch of the isotherm, giving rise to a hysteretic sorption behaviour typical for responsive MOFs exhibiting first-order phase transitions. Meanwhile, the Silicone 704 only has a very low $CO_2$ uptake of about 1.2 $cm^3$ (STP) $g^{-1}$ at 1224 mbar and 25 °C.

Strikingly, **PL7_10** and **PL7_5** also display sigmoidal $CO_2$ sorption isotherms with the eminent steps observed at ca. 650 mbar during adsorption and ca. 380 mbar during desorption. Of course, the overall uptake is higher for **PL7_10** (5.24 $cm^3 g^{-1}$) in comparison to **PL7_5** (3.12 $cm^3 g^{-1}$) due to the higher ZIF-7 nanocrystal fraction, which accounts for the majority of $CO_2$ uptake (Fig. 4b). Interestingly, a $CO_2$ sorption experiment performed without continuous stirring of **PL7_10** demonstrates that stirring is critical to overcome diffusion limitations through the Silicone 704. The $CO_2$ uptake of **PL7_10** at 1224 mbar is about 18% lower (4.45 $cm^3 g^{-1}$) when the sample is not stirred during the sorption experiment (Fig. 4b). Nevertheless, with stirring of the PLs, the $CO_2$ uptakes of **PL7_10** and **PL7_5** at a pressure of 1224 mbar approach closely the ideal values for both ZIF-7 suspensions calculated from the weighted contributions of the uptake of the PLs' pure constituents (Supplementary Notes—Gas sorption measurements)[14]. Specifically, the measured uptakes for **PL7_10** was 97% of the ideal value and the respective value for **PL7_5** corresponds to 94%, with the ZIF-7 nanocrystals being the main contributor to the overall $CO_2$ uptake of the PLs despite its low mass fraction (ca. 78% and ca. 66% of the total $CO_2$ uptake is attributed to the ZIF-7 nanocrystals for **PL7_10** and **PL7_5**, respectively). It is thus inferred that Silicone 704 has not

penetrated the pores of ZIF crystals. Furthermore, when the $CO_2$ isotherms of both PLs are normalized to the unit mass of ZIF-7, they very much resemble the isotherm of the pure solid (Fig. 4c), but exhibit a slight shift of the ***np-lp*** transition to higher $CO_2$ pressures, and similarly a slight shift of the ***lp-np*** transition to lower $CO_2$ pressures. These slight shifts of the transition pressures indicate moderate mass transport limitations through Silicone 704 matrix. However, performing the $CO_2$ gas sorption measurement of **PL7_10** with a longer equilibration time criterion (the time interval for the calculation of the pressure gradient has been increased from 12 to 18 min, the total time needed for collecting the isotherm increased from 19.7 to 29.1 h, see Supplementary Notes—Gas sorption measurements for details), does not lead to a narrowing of the hysteresis loop or a shift of the phase transition pressures towards those observed for the pure ZIF-7 solid (Supplementary Fig. 23). We anticipate that substantially longer equilibration times (presumably orders of magnitude longer) are required for the narrowing of the hysteresis to values similar to the solid phase.

The variation of the phase transition pressures, which are responsible for the position of the steps in the isotherms, with temperature, is of great importance for the application of such breathing PLs in temperature swing sorption processes, but also for the estimation of a suitable temperature range for pressure swing sorption processes[30,45]. In order to get a more detailed picture of the breathing behaviour of the responsive PLs, **PL7_10** was chosen to record $CO_2$ sorption isotherms at several temperatures in the range from 10 to 35 °C (all with continuous stirring of the PL during the data collection, Fig. 4d). As expected, the ***np-lp*** transition pressure during adsorption as well as the ***lp-np*** transition pressure during desorption decrease at a lower temperature and increase at a higher temperature. In particular, at 10 °C the ***np-lp*** transition pressure drops to ca. 370 mbar, while at 35 °C the transition starts at 880 mbar. Besides, the width of the hysteresis also diminishes with decreasing temperature, so that the most pronounced hysteretic behaviour is observed at 35 °C. Still, when the

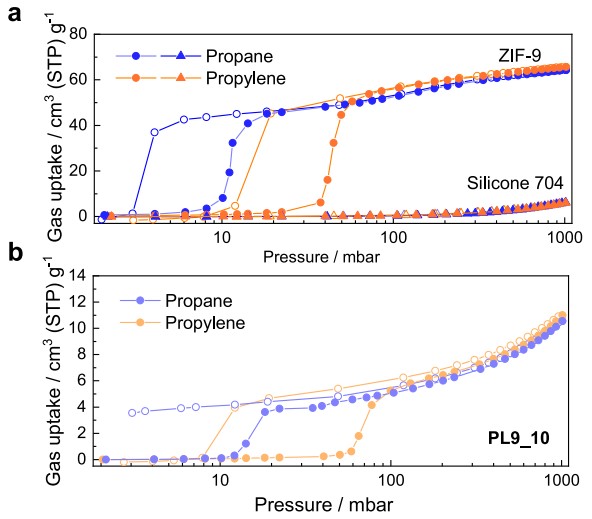

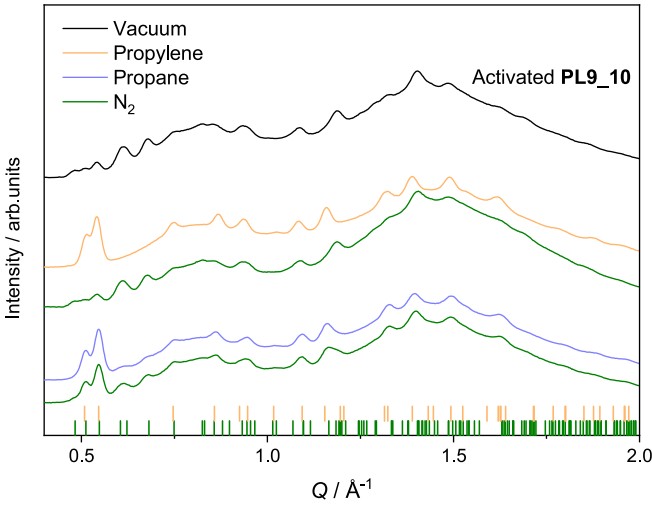

**Fig. 5 | Gas uptake behaviour of PL9 materials. a** Propane (blue) and propylene (orange) sorption isotherms of ZIF-9 nanoparticles (circles) and Silicone 704 (triangles) at 25 °C. **b** Propane (light purple) and propylene (light orange) sorption isotherms of **PL9_10** at 25 °C. For all isotherms, the adsorption branch is shown with filled symbols, and the desorption branch with empty symbols. Lines are a guide to the eye. **c** XRD patterns of activated **PL9_10** (black) and conditioned under -1 bar propylene (light orange) or $N_2$ (green) and 1 bar propane (light blue) or $N_2$. Tick marks represent the characteristic Bragg reflection positions of the **_lp_** phase of ZIF-9 (CCDC code VEJZEQ) and the **_np_** phase (CCDC code RIPNOV01) of ZIF-7. Only the lower Q region is displayed to highlight the differences. The full dataset of the collected patterns can be found in Supplementary Figures—In situ XRD.

isotherms are plotted on a relative pressure ($p/p_0$) scale ($p$ = absolute gas pressure, $p_0$ = saturation vapour pressure at the temperature of the data collection) very similar hysteresis widths are observed (Supplementary Fig. 28). A similar trend is observed for the pure ZIF-7 nanocrystals, in agreement with previous work (Supplementary Fig. 27)[46].

To further confirm that the gated $CO_2$ uptake of **PL7_10** and **PL7_5** is indeed due to the $CO_2$-responsive behaviour of the ZIF-7 nanocrystals, we prepared a related PL, denoted **PL8_5**, consisting of Silicone 704 with 5 wt% ZIF-8 nanocrystals (Zn(mim)$_2$; mim$^-$ = 2-methylimidazolate)[47]. Note that a PL of 10 wt% ZIF-8 in Silicone 704 turned out to be unstable, resulting in rapid sedimentation of the nanocrystals within ca. 1 h. No account of $CO_2$-induced flexibility in the examined pressure–temperature conditions have been given in the literature for ZIF-8[48] and hence no gated uptake was expected for the $CO_2$ sorption isotherm of **PL8_5**. Indeed, the $CO_2$ sorption isotherm of **PL8_5** at 25 °C displays a nearly linear shape without gated sorption, in agreement with the shape of the $CO_2$ sorption isotherm of the pure ZIF-8 crystals (see Supplementary Fig. 34).

Direct evidence for the $CO_2$-induced **_np_**–**_lp_** transition of the ZIF-7 nanocrystals as the cause of the sigmoidal sorption isotherms of **PL7_10** is obtained by in situ XRD experiments with synchrotron radiation. The activated **PL7_10** was placed into a cylindrical Schlenk tube (9 mm outer diameter) and an XRD pattern of the suspension was collected under vacuum. Subsequently, $CO_2$ and $N_2$ were alternately bubbled through the PL while XRD patterns were collected. The in situ XRD patterns, collected during three cycles of successive bubbling of $CO_2$ and $N_2$, establish the transition of the ZIF-7 nanocrystals contained in **PL7_10** from the **_np_** to the **_lp_** phase under $CO_2$ atmosphere and their return to **_np_** phase under the $N_2$ atmosphere (Fig. 4e). The reproducibility of the results in the 2nd and 3rd cycle reveals the full reversibility of the phase transition in the breathing PL system. Time-resolved XRD data reveal that the phase transitions immediately start after switching from $N_2$ to $CO_2$ (for the **_np_**–**_lp_** transition) or from $N_2$ to $CO_2$ (for the **_lp_**–**_np_** transition). In both cases, the transitions are complete after 5–15 min (Supplementary Figs. 37–39).

The $CO_2$ sorption performance of **PL9_10** was also studied. In agreement with the documented higher **_np_**–**_lp_** phase transition pressure of ZIF-9 compared to ZIF-7[35], the step in the $CO_2$ isotherm of

**PL9_10** starts close to 1100 mbar preventing us from observing the complete transition, as our gas sorption setup is limited to a maximum pressure of about 1224 mbar (Supplementary Figs. 30 and 31). Nevertheless, the data further evidence that the specific sorption behaviour of the responsive ZIF particles is fully preserved when they are suspended in Silicone 704, so that fine-tuning the position of the step in the isotherms of the breathing PLs could be straightforwardly performed by the concepts already established for the responsive MOF solids.

To unveil the potential of breathing PLs for utilization in applications with gases other than $CO_2$, we further investigated the breathing behaviour of **PL7_10** and **PL9_10** towards propane and propylene at 25 °C. Both, ZIF-7 and ZIF-9 solids, are known to also show breathing behaviour during the adsorption of C2 and C3 olefins and paraffins, which aroused avid interest for its application in hydrocarbon separations[37,38]. Indeed, both PLs feature characteristic steps in the sorption isotherms for both gases, on par with the corresponding isotherms of the pure solids. Similar to the $CO_2$ sorption behaviour, the **_np_**–**_lp_** phase transition of **PL7_10** during adsorption of propane and propylene occurs at a lower pressure compared to **PL9_10**. We focus the discussion here on **PL9_10**, while the corresponding sorption data of **PL7_10** are included in Supplementary Figures—Gas sorption measurements (Supplementary Fig. 33).

As a consequence of the much stronger interactions of ZIF-9 with propane and propylene compared to $CO_2$, the steps of their sorption isotherms are at a significantly lower pressure (Fig. 5a). During adsorption of the hydrocarbons in **PL9_10** the characteristic sigmoidal isotherm profile with a step at 15 mbar for propane and at 60 mbar for propylene is evident (Fig. 5b). Compared to the pure ZIF-9 nanocrystals, the steps in the isotherms of **PL9_10** occur at slightly higher gas pressures during adsorption and at slightly lower gas pressures during desorption, again indicating moderate mass transport limitations in the liquid phase. In fact, the very low **_lp_**–**_np_** phase transition pressure on the propane desorption branch of ZIF-9 (only 4 mbar for the pure nanocrystals) results in an open hysteresis loop for propane sorption in **PL9_10**; despite the strict equilibration criteria applied. This signifies that propane is strongly bound in the ZIF-9 particles and a rather high vacuum is required to desorb the gas from the ZIF-9 particles in **PL9_10** at 25 °C [38].

Similar to our previous observations for $CO_2$ sorption in **PL7_10**, in situ XRD studies of **PL9_10** under variable gas atmosphere testify the transition of the ZIF-9 nanocrystals in the PL from their ***np*** to their ***lp*** form when exposed to propylene or propane, and reversal to the ***np*** form when $N_2$ is bubbled through the PL (Fig. 5c, see Supplementary Methods−In situ XRD for details). Even though the transition to the ***lp*** phase upon hydrocarbon adsorption is fast (completed in < 6 min), the reverse transition to the ***np*** phase during $N_2$ bubbling is very slow in both cases. The full reversal to the ***np*** phase of the particles in the PL requires about 50 min of $N_2$ bubbling through the propylene-saturated **PL9_10** (Supplementary Fig. 40). Interestingly, bubbling of $N_2$ through the propane-saturated **PL9_10** did not secure the complete reversal of the ZIF-9 particles to their ***np*** phase (Supplementary Fig. 41). This agrees with the observed inability of the system to totally desorb propane even at a propane partial pressure of just 3 mbar (Fig. 4d). We anticipate that either a higher vacuum or a longer purging with inert gas is required to remove the propane from **PL9_10** and at the same time re-transform the ZIF-9 nanocrystals to the ***np*** form.

## Discussion

This work demonstrates MOF-based PLs which exhibit a breathing behaviour characterized by unique sigmoidal gas sorption isotherms for $CO_2$, propane and propylene. The combination of the breathing behaviour of the MOF particles with the fluidity of the liquid state, mimicking the complicated behaviour of complex biological fluids (i.e. blood), is exceptional for synthetic liquid systems. Advanced in situ XRD experiments verified that the sigmoidal gas sorption isotherms of the PLs are indeed driven by the gas-pressure-sensitive pore-volume-changing phase transitions of the MOF particles.

Generally, PLs are of the highest interest for applications in a wide range of fields including gas storage and adsorption, membrane separation, and catalysis. They are able to transcend limitations associated with the use of porous solids in such industrial applications owing to their distinctive fundamental advantage of fluidity together with their ability to be implemented in continuous processes. The breathing PLs introduced here achieve greater working capacities than conventional liquids or PLs based on non-responsive porous particles in similar pressure or temperature swings[32]. Characteristically, the formulated **PL7_10** displays superior uptake (4.1 cm$^3$ (STP) g$^{-1}$) compared to the commercial $CO_2$ sorbent Genosorb 1753 (2.33 cm$^3$ (STP) g$^{-1}$)[32] at 25 °C and 1 bar. Most importantly, in comparison with a high-performing PL based on rigid microporous organic particles with a similar weight fraction of the porous constituent, i.e. 12.5 wt% CC3-*R*/ CC3-*S* in silicone oil[32], **PL7_10** achieves a higher working capacity (3.85 cm$^3$ (STP) g$^{-1}$ compared to 3.32 cm$^3$ (STP) g$^{-1}$) for a 0.2–1 bar pressure swing, despite having a lower $CO_2$ uptake at 1 bar (4.14 cm$^3$ (STP) g$^{-1}$ compared to 4.44 cm$^3$ (STP) g$^{-1}$). The larger working capacity of **PL7_10** for such a pressure swing is a consequence of the PL's breathing behaviour and highlights the conceptual advantage of sigmoidal shape isotherms in gas separation applications. A table summarizing the $CO_2$ uptake capacities of PLs at ambient/sub-ambient conditions reported in the literature is provided in Supplementary Table 2. Unfortunately, the majority of the studies do not include desorption data, and hence we were not able to report working capacities, as the estimation of working capacity based solely on the adsorption branch can lead to largely overestimated values. Such differences can have a huge impact on the efficiency and economics of pressure and temperature swing adsorption processes, as regeneration conditions are a critical performance factor[49].

Compared to the recent advancements in the development of PLs based on non-responsive microporous particles, the $CO_2$ capacities of the breathing PLs reported here are relatively low[20,50]. Specifically, an uptake of ca. 16.53 cm$^3$ (STP) g$^{-1}$ was achieved by a 25 wt% zeolite RHO dispersion in Genosorb-1753[50]. The performance of a 5 wt% ZIF-8 dispersion in an IL is also impressive capturing 18.14 cm$^3$ (STP) g$^{-1}$[20].

Nevertheless, the new class of breathing PLs holds great potential and we are confident that significantly improved total capacities can be achieved by (i) increasing the weight fraction of the responsive MOF particles in the PL and (ii) employing other responsive MOF particles, which have higher gas capacities than ZIF-7 and ZIF-9.

To showcase the transferability of the presented concept to other responsive MOF particles, we include our first results on another breathing PL, which is based on a very different type of responsive MOF material. We prepared a literature-known amine-appended $Mg_2$(dobpdc) MOF (dobpdc$^{4-}$ = 4,4′-dioxidobiphenyl-3,3′-dicarboxylate)[51]. The amine-appended ii-2-$Mg_2$(dobpdc) material exhibits a higher $CO_2$ uptake capacity (77.7 cm$^3$ (STP) g$^{-1}$) at ambient conditions than ZIF-7 and also features a step-shaped sorption isotherm (ii-2 = N,N-diisopropylethylenediamine). The sigmoidal isotherm shape of this material, however, arises from the cooperative and reversible insertion of $CO_2$ into metal−amine bonds forming ammonium carbamate chains (i.e. a cooperative chemisorption process rather than physisorption)[51]. In this process, the volume of the MOF remains largely the same, so that the mechanism triggering the step-shaped $CO_2$ sorption isotherm is entirely different from the pore-volume-changing ***np***−***lp*** phase transition of ZIF-7 and ZIF-9. Indeed, a PL composed of only 5 wt% of ii-2-$Mg_2$(dobpdc) in Silicone 704 displays a step-shaped $CO_2$ sorption isotherm and features a $CO_2$ capacity of about 4.73 cm$^3$ (STP) g$^{-1}$ at 1.2 bar and 25 °C (Supplementary Fig. 35), which is quite impressive in view of the rather small weight fraction of the MOF in the PL (about 76% of the adsorbed $CO_2$ gas is hosted in the 5 wt% MOF present in the PL). The cyclic performance of the liquid system was also tested and the overall $CO_2$ capacity is only decreasing by 4% from the first to the second cycle, while it remains practically constant in the third cycle. Importantly, the sigmoidal shape of the sorption isotherm is preserved during cycling, even though the threshold pressure for $CO_2$ insertion (i.e. the inflection point of the isotherm) is varying slightly. Furthermore, the PL of ii-2-$Mg_2$(dobpdc) features high colloidal stability, as no sedimentation of the MOF particles is observed after 12 days of resting (Supplementary Fig. 17). This observation suggests that stable PLs with significantly higher MOF particle loadings and hence higher $CO_2$ uptakes could be developed based on amine-appended $Mg_2$(dobpdc).

Besides higher sorption capacities, future work should also focus on the development of breathing PLs with a narrower hysteresis width, in order to maximize the usable capacity by utilization of a minimal temperature or pressure swing. This can principally be accomplished by the employment of other responsive MOF particles exhibiting the desired properties[52,53]. We further consider exciting opportunities for the development of even more advanced responsive PLs by the implementation of other functional porous particles that exhibit responsiveness towards mechanical pressure[54], light[55], or maybe even electrical fields[56]. Indeed, during the writing of this paper, a light-responsive PL based on photosensitive MOF particles has been reported[33]. Such kind of multi-stimuli responsive liquid systems with remotely switchable porosity could set the stage for more efficient and sustainable processes, particularly in energy-related applications, such as molecular separation and capture technologies. This work thus highlights breathing PLs as a new family of sorbents, whose performance can be tuned from the materials perspective given the wide range of suitable constituents and numerous pairings of responsive porous nanocrystals with carrier liquids.

## Methods
### Materials syntheses
**ZIF-7.** 2724 mg $Zn(NO_3)_2$·$6H_2O$ was dissolved in 100 mL N,N-dimethylformamide (DMF), and 2400 mg benzimidazole (bimH) was dissolved in 100 mL methanol. After complete dissolution, the $Zn(NO_3)_2$·$6H_2O$/DMF solution was rapidly poured into the bimH/methanol solution. While stirring the mixture was heated at 110 °C

under reflux for ca. 4 h[39]. After cooling to room temperature, the as-synthesized ZIF-7 particles were collected by centrifugation ($7836 \times g$, 1.5 h).

**ZIF-8**. 1200 mg $Zn(NO_3)_2 \cdot 6H_2O$ was dissolved in 57.2 mL of methanol and 2640 mg of 2-methylimidazole (mimH) was dissolved in another 57.2 mL of methanol. After complete dissolution, the $Zn(NO_3)_2 \cdot 6H_2O$/methanol solution was poured into the mimH/solution solution. The mixture slowly became turbid and stirring continued for ca. 1 h at room temperature[47]. The as-synthesized ZIF-8 particles were collected by centrifugation ($7836 \times g$, 0.5 h).

**ZIF-9**. 1990 mg $Co(CH_3COO)_2 \cdot 4H_2O$ was dissolved in 80 mL methanol and 1800 mg bimH was dissolved in 80 mL DMF. After complete dissolution, the $Co(CH_3COO)_2 \cdot 4H_2O$/MeOH solution was poured into the bimH/DMF solution quickly. The mixture was stirred at room temperature for ca. 5 h[40]. The as-synthesized ZIF-9 particles were collected by centrifugation ($7836 \times g$, 1.5 h).

**Workup of ZIF nanocrystals**. After collecting the ZIF particles by centrifugation, they were washed 3 times with ca. 40 mL methanol to remove unreacted starting materials and byproducts. Afterward, the nanocrystal powders were dried under air at 60 °C for 2 h and activated overnight at 180 °C under a dynamic vacuum (approx. $10^{-2}$ mbar).

**ii-2-Mg$_2$(dobpdc)**. The framework $Mg_2$(dobpdc) was synthesized by a solvothermal method scaled down from a previous report[51]. The ligand $H_4$(dobpdc) (4,4'-dihydroxy-[1,1'-biphenyl]-3,3'-dicarboxylic acid, 711 mg) and $Mg(NO_3)_2 \cdot 6H_2O$ (858 mg) were mixed in 15 mL of a 55:45 (v:v) methanol:DMF mixture and sonicated until complete dissolution. The solution was added to a 100 mL Schott vessel sealed with a high temperature–pressure polybutylene terephthalate cap and heated for 24 h in an oven at 120 °C. The crude white powder was isolated by filtration and washed three times in 50 mL of DMF, followed by solvent exchange by soaking three times in 50 mL of methanol. Subsequently, approximately 120 mg of methanol-solvated $Mg_2$(dobpdc) was isolated by filtration. After collection of a PXRD pattern, the $Mg_2$(dobpdc) was washed with 30 mL of toluene, and submerged in 20 mL of a 20% (v/v) solution of *N,N*-diisopropylethylenediamine in toluene. After 24 h, the solid was isolated by filtration and washed with fresh toluene to remove excess diamine and to obtain ii-2-$Mg_2$(dobpdc).

**PL7/PL8/PL9**. The activated ZIF powders (75 mg for 5 wt% dispersions and 150 mg for 10 wt% dispersions) were dispersed in Silicone 704 (1425 mg for 5 wt% and 1350 mg for 10 wt% dispersions) and sonicated for ca. 15 min until the formation of a homogeneous suspension was noticed.

**PL-ii-2-Mg$_2$(dobpdc)**. After heating for 12 h at 120 °C under vacuum, 53 mg of the activated ii-2-$Mg_2$(dobpdc) were dispersed in Silicone 704 (1013 mg) and sonicated for ca. 15 min to obtain a homogeneous suspension of 5 wt% ii-2-$Mg_2$(dobpdc) in Silicone 704.

### Gas sorption measurements
Sorption experiments were undertaken with a Quantachrome Autosorb iQ MP porosimeter using only high-purity adsorptive gases ($CO_2$: 99.995%, propane: 99.95%, propylene: 99.95%). Sample quantities of at least 40 mg were used for the experiments of the ZIF nanocrystals. Prior to the measurements solid ZIF samples were carefully ground and subsequently degassed in a dynamic vacuum ($p \approx 10^{-4}$ mbar) at 180 °C. In the case of $Mg_2$(dobpdc) and ii-2-$Mg_2$(dobpdc), the samples were degassed at 120 °C. A filler rod was used with the sample cells in order to reduce the dead volume. The analysis temperature was controlled with a Julabo CORIO CD-200F circulator. The porosimeter was modified to allow the conduction of sorption measurements of liquid

samples under continuous stirring. Further information on the setup and the equilibration criteria is provided in Supplementary Methods-Gas sorption measurements and Supplementary Notes-Gas sorption measurements. Prior to analysis, the PLs were degassed at 100 °C for 6 h under dynamic vacuum ($p \approx 10^{-4}$ mbar), apart from cycle 3 of the $CO_2$ sorption analysis of PL-ii-$Mg_2$(dobpdc) where the activation temperature was 120 °C. The weight of the liquid samples loaded into the cells (0.45–1.06 g) was measured before and after degassing. A Kern ABJ 220-4NM analytical balance with reproducibility of 0.2 mg was used. No weight loss higher than 1 mg was noticed, in line with the low vapour pressure of the Silicone 704 at the measurement temperature ($10^{-9}$ mbar).

### Viscosity measurements
An MCR502 Rheometer from Anton Paar with a plate-plate-geometry (plate diameter 50 mm) was used. The distance between the plates was 0.5–0.6 mm. The investigated range of shear rate was 0.1–1000 s$^{-1}$. A cooling unit Evo 20 was used for temperature control at 25 and −10 °C.

### TGA analysis
Thermogravimetric analysis was performed with a TA Instruments SDT-650 instrument using alumina cups and a 10 °C min$^{-1}$ heating rate up to 600 °C in nitrogen atmosphere ($N_2$ flow of ~100 mL/min). In the case of $Mg_2$(dobpdc) and ii-2-$Mg_2$(dobpdc) a heating rate of 3 °C min$^{-1}$ was used with an upper temperature bound of 500 °C.

### Powder X-ray diffraction
Powder X-ray diffraction data of as-synthesized and activated materials were collected on a Siemens D5005 diffractometer in Bragg–Brentano geometry using CuKα ($\lambda = 1.5418$ Å) radiation in a 2θ range from 5° to 50° with a step size of 0.02°. All samples were finely ground and placed on a glass holder. Sample purity was confirmed by profile fits (Pawley method[57]) using TOPAS academic v6 software package[58].

### Scanning electron microscopy
Scanning electron microscopy (SEM) was performed with a Hitachi S-4500 instrument with an acceleration voltage of 1.0 kV. The powders were deposited on a conductive carbon pad.

### Transmission electron microscopy
Transmission electron microscopy images were recorded on a Thermo Fisher Talos-F200X applying a 200 kV acceleration voltage. Powdered samples were deposited on a carbon-coated TEM grid.

### In situ XRD
Collection of X-ray diffraction data of the PLs under $CO_2$, propane and propylene sorption was conducted at beamline BL9 at DELTA (Dortmunder Elektronenspeicherring-Anlage, Dortmund, Germany) with a monochromatic X-ray beam ($\lambda = 0.45904$ Å) using a MAR345 image plate detector[59]. 1.5 mL of a PL was loaded in a glass Schlenk tube (9 mm diameter) and the sample was mounted over a magnetic stirring plate in the centre of the goniometer at BL9. The Schlenk tube was sealed with a septum and the tube's sidearm served as the gas outlet. The gas was bubbled through the PL via a needle pierced through the septum and connected to the tubing. The tubing was connected to a three-way ball valve to switch between sorption gas ($CO_2$, propane or propylene) and purge gas ($N_2$) while collecting diffraction patterns. Photos of the experimental setup are provided in Supplementary Fig. 2.

### Dynamic light scattering (DLS)
DLS measurements were performed using a Malvern Zetasizer Nano-ZS instrument. **PL7_10** was diluted with Silicone 704 to obtain a final concentration of about 75 μg mL$^{-1}$. It should be noted that dilution is required to obtain accurate DLS data for nanoparticle suspensions

with high concentrations. Six replicate measurements were carried out. The measurement temperature was 25 °C, the refractive index for the particles and the dispersion medium were 1.47[60] and 1.4 (according to the sample dispersion and refractive index guide of Malvern Instruments master-sizer 2000 reference manual), respectively, and the viscosity of the medium was 42 mPa s (in agreement with the viscosity measurement).

### ¹H NMR spectroscopy

Approximately 10 mg of powder was digested in a solution of 0.1 mL of 35 wt.% DCl in $D_2O$ and 0.6 mL of DMSO-$d_6$. ¹H NMR spectra were acquired on Bruker DPX300, DPX 500 or Agilent DD2 500 spectrometers.

## Data availability

The authors declare that all data supporting the findings of this study are available within the article and its supplementary information files. The corresponding raw data are available on request from the corresponding author S.H.

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

## Acknowledgements

This project was funded within the programme "Experiment!" of the Volkswagen Foundation (grant number A131005). LFB acknowledges the Priority Programme SPP1928 COORNETs for a Gateway Fellowship. We further acknowledge the DELTA machine staff for support during our synchrotron XRD experiments, Volker Brand for the operation of SEM and TEM instruments, Philipp Muenzner for contributing to the viscosity measurements and Dr. Suresh Vasa for his guidance for DLS analysis. Andrea Machalica is acknowledged for performing preliminary work on this project. We acknowledge financial support by Deutsche Forschungsgemeinschaft and Technische Universität Dortmund/TU Dortmund University within the funding programme Open Access Costs.

## Author contributions

A.K. synthesized the materials and performed all experiments as well as data analysis. R.P., L.F.-B., C.D., M.P. and C.S. contributed to the in situ XRD experiments of the PLs at DELTA. A.K. and S.H. wrote the manuscript with contributions from all the authors. All authors have given approval of the final manuscript.

## Funding

## Competing interests

The authors declare no competing interests.
