## [Peer Review File · Nature Communications]

REVIEWER COMMENTS

Reviewer #1 (Remarks to the Author):

This manuscript presents porous liquids with breathing behaviours based on two ZIFs dispersed in silicone oil. The formed porous liquids show gated CO₂, propane and propylene uptakes as expected. There are interesting results in the manuscript. However, the gas adsorption behaviour of Type 3 porous liquids is usually dominated by solid components. The gated adsorption behaviour of PL7 and PL9 can be almost predicted based on ZIF-7 and ZIF-9, since there are already many papers published about the gated adsorption behaviour of ZIF-7 and ZIF-9. The author also used in situ XRD to study the gas-pressure-triggered phase transition of the ZIFs in liquid dispersions. However, the phase transition of ZIF-7 has been studied using in situ XRD by Xiang, L. et al. (Locking of Phase Transition in MOF ZIF-7: Improved Selectivity in Mixed-Matrix Membranes for O₂/N₂ Separation. *Mater. Horiz.* 2020, 7 (1), 223–228). Furthermore, it is not the first time that such PL systems were reported as the author mentioned in the manuscript. Although the phase transition of PLs was studied, the results are consistent with Xiang's work. Therefore, due to the lack of novelty, this work is not suggested to publish in *Nature Communications*.

Reviewer #2 (Remarks to the Author):

The revised manuscript by Henke and coworkers addresses the majority of my previous comments, and I appreciate the effort made to perform more experiments and include another material in their work. However, some concerns remain, particularly with respect to characterization of the newly included material ii-2-Mg₂(dobpdc). Still, I believe the revised manuscript is impactful and suitable for publication in *Nat. Commun.* after revisions are made to address the points below.

- The addition of the material ii-2-Mg₂(dobpdc) broadens the scope and further supports the claims of this paper. However, this material did not receive the same level of characterization that the ZIF-7 and ZIF-9 dispersions did. At a minimum, the quality of the as-synthesized ii-2-Mg₂(dobpdc) should be established, and ideally, the material should be characterized after dispersion to establish that it is stable in silicone 704.

- o The diamine loading is a fair bit lower than reported in the literature (e.g. the paper cited by the authors, <https://doi.org/10.1021/jacs.7b05858>, reports a loading of 103% vs the authors' 77%). Could the authors explain this and its possible effects on the capacity and performance of their dispersions?

- o As this is the first known instance of using ii-2-Mg₂(dobpdc) for CO₂ adsorption in silicone oil, could the authors also provide insights into the stability of the material? In particular, providing solid-state CO₂ capacity of ii-2-Mg₂(dobpdc) before and after measurement would support the claims of material stability and ensure that there is no amine loss upon prolonged exposure to oil.

- o There is no mention of the particle size or colloidal stability of the ii-2-Mg₂(dobpdc) dispersions, which is surprising given that the authors discuss the importance of particle size for ZIF-7 and ZIF-9 dispersions. As far as this reviewer is aware, ii-2-Mg₂(dobpdc) has not been made in nanoparticle form, so clarifying this point is important broadly for the field.

- On pg 10, line 9-10, the authors compare their porous liquid samples to the commercial solvent Genosorb at 25 °C and 1 bar. However, Genosorb is typically used at much higher pressures (> 5 bar), so the comparison is not completely representative of how these materials would be used.

- The scale bars for the SEM images in Figure S8 difficult to read and would benefit from being made larger in size.

Reviewer #3 (Remarks to the Author):

The authors have thoroughly responded to all the comments made by all three reviewers, including adding lengthy passages to the manuscript to cover points made, and carried out further experiments enhancing the findings already in the original submission.

The only comment from this reviewer is regarding the response to reviewer #2 point 8 and additional experiments carried out for this. The authors postulate that the widening of the hysteresis loop in PL7_10 and PL7_5 compared to solid ZIF-7 is due to kinetic limitations of gas adsorption due to reduced mass transfer in the porous liquids. The authors show (manuscript page 7 line 6) that with extending the equilibration times of the gas sorption experiments, this hysteresis loop is unchanged. The authors should further comment on this in the manuscript as this does not agree with their hypothesis following on from their reviewer response or the point made in the manuscript - it is important to state in the main text that the increase of equilibration time was only 1.48 ($\Delta t = 12$ minutes increased to 18 minutes), and that significantly longer equilibration times required are not possible (capability of Quantachrome autosorb iQ MP is not known by the reviewer, other Quantachrome instruments can run equilibration times of up to 30 minutes, and the authors show use of 24 minutes in their response). The authors should state they expect many orders of magnitude greater equilibrations to observe the expected narrowing in the main text. Using an equilibration time increase of 2-3 rather than 1.48 could also be sufficient to observe narrowing of the hysteresis.

Otherwise, this reviewer is satisfied by the response and changes made by the authors and thanks them for their response, and believes this work to be suitable for publication in Nature Communications.

Reviewer #4 (Remarks to the Author):

The authors have thoroughly responded to all the comments made by all three reviewers, including adding lengthy passages to the manuscript to cover points made, and carried out further experiments enhancing the findings already in the original submission.

The only comment from this reviewer is regarding the response to reviewer #2 point 8 and additional experiments carried out for this. The authors postulate that the widening of the hysteresis loop in PL7_10 and PL7_5 compared to solid ZIF-7 is due to kinetic limitations of gas adsorption due to reduced mass transfer in the porous liquids. The authors show (manuscript page 7 line 6) that with extending the equilibration times of the gas sorption experiments, this hysteresis loop is unchanged. The authors should further comment on this in the manuscript as this does not agree with their hypothesis following on from their reviewer response or the point made in the manuscript - it is important to state in the main text that the increase of equilibration time was only 1.48 ($\Delta t = 12$ minutes increased to 18 minutes), and that significantly longer equilibration times required are not possible (capability of Quantachrome autosorb iQ MP is not known by the reviewer, other Quantachrome instruments can run equilibration times of up to 30 minutes, and the authors show use of 24 minutes in their response). The authors should state they expect many orders of magnitude greater equilibrations to observe the expected narrowing in the main text. Using an equilibration time increase of 2-3 rather than 1.48 could also be sufficient to observe narrowing of the hysteresis.

Otherwise, this reviewer is satisfied by the response and changes made by the authors and thanks them for their response, and believes this work to be suitable for publication in Nature Communications.

RESPONSE TO REVIEWERS' COMMENTS

A point-by-point response to the reviewers follows. Figure, page and line number refer to the updated versions of the manuscript and the Supplementary Information. The changes within the manuscript and the Supplementary Information have been highlighted in yellow.

Reviewer #1 (Remarks to the Author):

This manuscript presents porous liquids with breathing behaviours based on two ZIFs dispersed in silicone oil. The formed porous liquids show gated CO₂, propane and propylene uptakes as expected. There are interesting results in the manuscript. However, the gas adsorption behaviour of Type 3 porous liquids is usually dominated by solid components. The gated adsorption behaviour of PL7 and PL9 can be almost predicted based on ZIF-7 and ZIF-9, since there are already many papers published about the gated adsorption behaviour of ZIF-7 and ZIF-9. The author also used in situ XRD to study the gas-pressure-triggered phase transition of the ZIFs in liquid dispersions. However, the phase transition of ZIF-7 has been studied using in situ XRD by Xiang, L. et al. (Locking of Phase Transition in MOF ZIF-7: Improved Selectivity in Mixed-Matrix Membranes for O₂/N₂ Separation. Mater. Horiz. 2020, 7 (1), 223–228). Furthermore, it is not the first time that such PL systems were reported as the author mentioned in the manuscript. Although the phase transition of PLs was studied, the results are consistent with Xiang's work. Therefore, due to the lack of novelty, this work is not suggested to publish in Nature Communications.

Response

We thank the reviewer for their critical feedback. We regret that the reviewer, in contrast to the other three reviewers, does not appreciate the novelty and impact of our work. We would like to highlight the most important findings of our work again and sincerely hope that the reviewer agrees that our revised manuscript is of high relevance for the field of porous liquids:

- Even though a breathing PL based on ZIF-7 and sesame oil has been reported previously, we provide the first in-depth study on the sorption properties of a ZIF-7-based breathing PL including high-resolution gas sorption isotherms, also providing sorption data at variable temperatures. Since the phase transition pressure of responsive MOFs is a function of the temperature, the temperature-dependent sorption behaviour of the corresponding breathing PLs is of high relevance.
- We report first sorption data on a related breathing PL containing the isostructural cobalt-based material ZIF-9.
- We report the first *in situ* XRD study of PLs and provide first experimental evidence for the phase transformation of dispersed MOF particles during gas sorption in a liquid. We show this for various gases and we demonstrated cyclability of the breathing PLs.
- The *in situ* XRD data also gives first insights into the important kinetics of the phase transformations.
- We now include a significantly expanded analysis of a very new type of breathing PL based on the totally different amine-appended Mg₂(dobpdc) system. This significantly broadens the scope of our approach and sets the ground for the development of a series of highly tailorable PLs with sigmoidal gas isotherms.

We strongly believe that our work is a very significant addition, likely inspiring other research to make further new discoveries in this exciting field of porous materials research.

Reviewer #2 (Remarks to the Author):

The revised manuscript by Henke and coworkers addresses the majority of my previous comments, and I appreciate the effort made to perform more experiments and include another material in their work. However, some concerns remain, particularly with respect to characterization of the newly

included material ii-2-Mg₂(dobpdc). Still, I believe the revised manuscript is impactful and suitable for publication in Nat. Commun. after revisions are made to address the points below.

- The addition of the material ii-2-Mg₂(dobpdc) broadens the scope and further supports the claims of this paper. However, this material did not receive the same level of characterization that the ZIF-7 and ZIF-9 dispersions did. At a minimum, the quality of the as-synthesized ii-2-Mg₂(dobpdc) should be established, and ideally, the material should be characterized after dispersion to establish that it is stable in silicone 704.
 - o The diamine loading is a fair bit lower than reported in the literature (e.g. the paper cited by the authors, <https://doi.org/10.1021/jacs.7b05858>, reports a loading of 103% vs the authors' 77%). Could the authors explain this and its possible effects on the capacity and performance of their dispersions?
 - o As this is the first known instance of using ii-2-Mg₂(dobpdc) for CO₂ adsorption in silicone oil, could the authors also provide insights into the stability of the material? In particular, providing solid-state CO₂ capacity of ii-2-Mg₂(dobpdc) before and after measurement would support the claims of material stability and ensure that there is no amine loss upon prolonged exposure to oil.
 - o There is no mention of the particle size or colloidal stability of the ii-2-Mg₂(dobpdc) dispersions, which is surprising given that the authors discuss the importance of particle size for ZIF-7 and ZIF-9 dispersions. As far as this reviewer is aware, ii-2-Mg₂(dobpdc) has not been made in nanoparticle form, so clarifying this point is important broadly for the field.

Response:

We thank the reviewer for the feedback provided and the suggestions made to improve our manuscript.

- Indeed, the fraction of diamine loading likely plays an important role for the ambient pressure CO₂ capacity of ii-2-Mg₂(dobpdc) (expected to increase with the diamine loading) as well as the threshold pressure for CO₂ insertion (we assume this is decreasing with amine loading). In order to better compare the behaviour of our ii-2-Mg₂(dobpdc) material and its PL with data from the literature, we have now prepared another batch of ii-2-Mg₂(dobpdc) and conducted further analyses. In comparison to the synthetic steps followed for the original ii-2-Mg₂(dobpdc) synthesis, we have this time submerged the Mg₂(dobpdc) framework in the 20% (v/v) solution of *N,N*-diisopropylethylenediamine in toluene for 24 h (instead of 12 h) under mild stirring. We were thus able to obtain a diamine loading of 94% (according to ¹H NMR data, Figure S15). In terms of CO₂ capacity, the newly synthesised ii-2-Mg₂(dobpdc) displays a CO₂ uptake of 78.78 cm³ (STP) g⁻¹ at 25 °C and 1128 mbar (see below), while the respective value for the reported material is ca. 76 cm³ (STP) g⁻¹ (see <https://doi.org/10.1021/jacs.7b05858>). In both cases the “step” in the CO₂ isotherm is observed in the range of 250-300 mbar. The new batch of ii-2-Mg₂(dobpdc) has also been characterised by means of PXRD and TGA and the relevant graphs (Figures S7 and S21) have now been updated. All data from the previous ii-2-Mg₂(dobpdc) sample having 77% of diamine loading have been removed from the manuscript and the supplementary information. We now only discuss the new material featuring 94% of diamine loading.
- After its synthesis, ii-2-Mg₂(dobpdc) was degassed overnight at 120 °C under vacuum and formulated into a 5 wt% PL. Specifically, Silicone 704 was added and the dispersion was sonicated for 10 min giving rise to a milky liquid (Figure S17). The obtained PL was then tested for its CO₂ capacity (equilibrium criteria applied for **PL7** and **PL9** were used) and achieved the remarkable capacity of 4.73 cm³ (STP) g⁻¹, corresponding to the 93% of the ideal uptake.
Unfortunately, the method for the evaluation of the stability of ii-2-Mg₂(dobpdc) in Silicone 704 proposed by the reviewer, i.e. measurement of the solid phase CO₂ capacity before and after dispersion in Silicone 704, is not applicable for the case of PL-ii-2-Mg₂(dobpdc). Due to the very low vapour pressure of Silicone 704, a temperature well above 250 °C (see Figure S18) would be needed for the complete evaporation of Silicone 704 from the sample. This

temperature is too high and would cause (at least partial) decomposition of ii-2-Mg₂(dobpdc) so that the sorption capacity of the sample cannot be determined anymore (see Figure S21). Thus, we settled on a different approach to testify the stability of PL-ii-2-Mg₂(dobpdc). Three consecutive CO₂ adsorption/desorption isotherms were recorded for PL-ii-2-Mg₂(dobpdc) to provide insights into both, PL stability and cyclability. After the collection of the first CO₂ isotherm, the PL-ii-2-Mg₂(dobpdc) was stored for 4 days at ambient conditions and a second adsorption/desorption cycle was run after degassing at 100 °C. Subsequently, a third cycle was run with a regeneration temperature of 120 °C being used this time. The results are now provided in Figure S35 (see below). Only a slight reduction of the equilibrium uptake and a small shift of the isotherm step were observed with increasing cycle number. This observation indicates the stability of the framework after its dispersion in Silicone 704.

The following text has been now added on page 10 of the manuscript:

Indeed, a PL composed of only 5 wt% of the amine-appended Mg₂(dobpdc) in Silicone 704 displays a step-shaped CO₂ sorption isotherm and features a CO₂ capacity of about 4.73 cm³ (STP) g⁻¹ at 1.2 bar and 25 °C (Figure S35), which is quite impressive in view of the rather small weight fraction of the MOF in the PL (about 76% of the adsorbed CO₂ gas is hosted in the 5 wt% MOF present in the PL). The cyclic performance of the liquid system was also tested and the overall CO₂ capacity is only decreasing by 4% from the first to the second cycle, while it remains practically constant in the third cycle. Importantly, the sigmoidal shape of the sorption isotherm is preserved during cycling, even though the threshold pressure for CO₂ insertion (i.e. the inflection point of the isotherm) is varying slightly. Furthermore, the PL of ii-2-Mg₂(dobpdc) features a high colloidal stability, as no sedimentation of the ii-2-Mg₂(dobpdc) particles is observed after 12 days of resting (Figure S17). This observation suggests that stable PLs with significantly higher MOF particle loadings and hence higher CO₂ uptakes could be developed based on amine-appended Mg₂(dobpdc).

The following image is included in the Supplementary Information:

Figure S17: Photographs of PL-ii-2-Mg₂(dobpdc) (5 wt% ii-2-Mg₂(dobpdc) in Silicone 704) directly after synthesis (left), after 12 days of storage (right). The sample was stored at ambient conditions without any agitation. Sedimentation of the MOF particles is not observable. The slight colour changes in PL are due to different lighting conditions in the laboratory.

Figure S35: CO₂ sorption isotherms of ii-2-Mg₂(dobpdc) (top) and its respective 5wt% PL in Silicone 704 (bottom) at 25 °C. For all isotherms the adsorption branch is shown with filled symbols and the desorption branch with empty symbols, while the lines are a guide to the eye. Similar to the observations for the **PL7** and **PL9** materials, the step during adsorption is shifted to slightly higher gas pressures, whereas the step during desorption is shifted to slightly lower gas pressures. The uptake of the PL at 1224 mbar is in good agreement with the uptake expected from the constituents, with the experimental value being equal to 93% of the ideal uptake. It is estimated that the Silicone 704 contributes a CO₂ uptake of about 1.13 cm³ (STP) g⁻¹ PL, while the ii-2-Mg₂(dobpdc) particles contribute about 3.60 cm³ (STP) g⁻¹ PL. This is quite impressive, since the 5 wt% ii-2-Mg₂(dobpdc) particles in the PL contribute about 76% of the CO₂ uptake capacity of the PL at 1224 mbar. Three adsorption/desorption cycles were carried out with the PL being regenerated under vacuum at 100 °C and 120 °C, before cycle 2 and cycle 3, respectively. Between cycles 1 and 2 the sample rested for 4 days at ambient conditions.

- SEM analysis has now been carried out to provide information on the morphology of the Mg₂(dobpdc) and ii-2-Mg₂(dobpdc). The obtained images are provided in Figure S9 (see below). As expected, conventional solvothermal synthesis gave rise to high aspect-ratio microrods. The microrods of Mg₂(dobpdc) and ii-2-Mg₂(dobpdc) look practically identical confirming the successful grafting of ii-2 into the pores of the MOF without any morphological changes of the framework. Despite the relatively large crystal size of ii-2-Mg₂(dobpdc), the 5 wt% formulation appears to be highly stable as no signs of sedimentation were observed after 12 days of resting without agitation. We are thus very confident that PLs with significantly higher Mg₂(dobpdc) weight fractions can be developed and higher CO₂ uptake capacities can be achieved. Besides, recent studies developed synthetic routes for the scaling of ii-2-Mg₂(dobpdc) to nanoscopic dimensions which could potentially allow weight fractions far in excess of 10 wt% (<https://doi.org/10.1021/acs.nanolett.7b03106>). We are currently working in this direction and we hope to be able to report breathing PLs with very high working capacities in the future.

The following figure is included in the Supplementary Information:

Mg₂(dobpdc)

ii-2-Mg₂(dobpdc)

Figure S9: SEM micrographs of Mg₂(dobpdc) and ii-2-Mg₂(dobpdc) crystals used for the formulation of respective PLs.

- On pg 10, line 9-10, the authors compare their porous liquid samples to the commercial solvent Genosorb at 25 °C and 1 bar. However, Genosorb is typically used at much higher pressures (> 5 bar), so the comparison is not completely representative of how these materials would be used.

Response

Unfortunately, we are unable to compare the sorption properties of our breathing PLs with Genosorb at CO₂ pressures higher than around 1.2 bar, since we are unable to collect high pressure gas sorption isotherms in our lab. We agree with the reviewer that Genosorb is typically applied at higher gas pressures, for example in biogas upgrading. The main reason is that the performance of Genosorb is not economical at lower pressures (see for example <https://doi.org/10.3389/fenrg.2020.560849>). However, one has to note that the partial pressure of CO₂ is only about 1.5 bar, when Genosorb is applied in biogas upgrading at a total gas pressure of 5 bar (if we assume a mole fraction of 30% CO₂ in biogas, see <https://doi.org/10.1021/acs.est.9b03003>). Furthermore, a high CO₂ capacity at low CO₂ partial pressures is important for the efficiency of a material in biogas upgrading, as this leads to a low mole fraction (and thus a low partial pressure) of CO₂ in the upgraded gas (i.e. high purity methane from biogas). Hence, the physical parameters applied in our comparison (25 °C and 1 bar CO₂ pressure) are still of relevance for such technological processes.

We would like to stress that our manuscript does not aim at a technoeconomic assessment of our PLs with respect to commercial technologies. Our goal is to provide a first proof of concept that breathing PLs can be prepared based on two very different responsive MOF systems and that these materials feature immense potential for applications in separation technologies.

- The scale bars for the SEM images in Figure S8 difficult to read and would benefit from being made larger in size.

Response

Thank you for this suggestion. The images in Figure S8 are now displayed with a larger size so that the relevant scale bars are easier to be read.

Reviewer #3 and #4 (Remarks to the Author):

The authors have thoroughly responded to all the comments made by all three reviewers, including adding lengthy passages to the manuscript to cover points made, and carried out further experiments enhancing the findings already in the original submission.

The only comment from this reviewer is regarding the response to reviewer #2 point 8 and additional experiments carried out for this. The authors postulate that the widening of the hysteresis loop in PL7_10 and PL7_5 compared to solid ZIF-7 is due to kinetic limitations of gas adsorption due to reduced mass transfer in the porous liquids. The authors show (manuscript page 7 line 6) that with extending the equilibration times of the gas sorption experiments, this hysteresis loop is unchanged. The authors should further comment on this in the manuscript as this does not agree with their hypothesis following on from their reviewer response or the point made in the manuscript - it is important to state in the main text that the increase of equilibration time was only 1.48 ($\Delta t = 12$ minutes increased to 18 minutes), and that significantly longer equilibration times required are not possible (capability of Quantachrome autosorb iQ MP is not known by the reviewer, other Quantachrome instruments can run equilibration times of up to 30 minutes, and the authors show use of 24 minutes in their response). The authors should state they expect many orders of magnitude greater equilibrations to observe the expected narrowing in the main text. Using an equilibration time increase of 2-3 rather than 1.48 could also be sufficient to observe narrowing of the hysteresis.

Otherwise, this reviewer is satisfied by the response and changes made by the authors and thanks them for their response, and believes this work to be suitable for publication in Nature Communications.

Response

We thank the reviewer for the time and effort to re-evaluate our manuscript. We totally agree with the comment regarding the hysteresis broadening in liquid systems in comparison to their solid counterparts. The fact that an increase of the total experiment time by 50% does not lead to any

significant changes in the sorption isotherm (capacity, phase transition pressure, hysteresis width) strongly suggests (several) orders of magnitude longer equilibration times would be required to observe the expected narrowing. Hence, such experiments would be extremely time demanding and would also not aid the assessment of the technological potential of these breathing PLs.

In order to make this clearer, we have now modified the text in Page 7 lines 9-15 as follows:

*However, performing the CO₂ gas sorption measurement of **PL7_10** with a longer equilibration time criterium (the time interval for the calculation of the pressure gradient has been increased from 12 min to 18 min, the total time needed for collecting the isotherm increased from 19.7 h to 29.1 h, see Supplementary Information for details), does not lead to a narrowing of the hysteresis loop or a shift of the phase transition pressures towards those observed for the pure ZIF-7 solid (Figure S23). We anticipate that substantially longer equilibration times (presumably orders of magnitude longer) are required for the narrowing of the hysteresis to values similar to the solid phase.*

REVIEWERS' COMMENTS

Reviewer #1 (Remarks to the Author):

I believe that the authors have in most cases responded well to the comments of the referee.

An outstanding issue remains the level of significance ascribed to the previous demonstration of the concept on which this paper is based, i.e. reference 13.

"Previously, Lai et al. observed a step- 3 1 shaped ethene sorption isotherm for a sesame-oil-based PL containing ZIF-7 particles (ZIF-7 = $Zn(bim)_2$, bim- = benzimidazolate, ZIF = zeolitic imidazolate framework). 13 2 The stepped shape 3 isotherm was rationally attributed to the gas-pressure-responsive behaviour of the ZIF-7 particles, 4 however, only very few data points have been recorded in the step region of the isotherm and the origin 5 of the step was not investigated experimentally. Band the origin 5 of the step was not investigated experimentally"

This statement is disingenuous since the origin of the step behaviour has previously been well characterised in the solid state. The authors must reword appropriately.

"Moreover, the previous study by Lai et al. already suggested that at least ZIF-7 is a suitable material for the development of breathing PLs.13"

Again this is disingenuous because the Lai work clearly showed that step-changes can be incorporated into PLs, it was not just "suggested". The authors must reword appropriately.

Following these changes I will recommend publication.

Reviewer #2 (Remarks to the Author):

The authors did an excellent job of addressing my previous comments. Though the novelty of this work is tempered by a previous study of ZIF-7 dispersions, I think there is sufficient novelty here to warrant publication in Nature Communications.

Reviewer #4 (Remarks to the Author):

Following on from the most recent round of peer review, this reviewer believes that the authors have addressed all of the reviewer comments, the science is sound and thorough, including additional experimental evidence.

RESPONSE TO REVIEWERS' COMMENTS

A point-by-point response to the reviewers follows. Figure, page and line number refer to the updated versions of the manuscript and the Supplementary Information. The changes within the manuscript have been highlighted in yellow.

Reviewer #1 (Remarks to the Author):

I believe that the authors have in most cases responded well to the comments of the referee.

Response

We are happy that the reviewer is overall pleased with our response to the previous comments. The following changes were applied to address their concerns over attribution to the previous work by Lai et al. (<https://doi.org/10.1021/acsami.0c19044>):

An outstanding issue remains the level of significance ascribed to the previous demonstration of the concept on which this paper is based, i.e. reference 13.

- *Previously, Lai et al. observed a step- 3 1 shaped ethene sorption isotherm for a sesame-oil-based PL containing ZIF-7 particles (ZIF-7 = Zn(bim)₂, bim⁻ = benzimidazolate, ZIF = zeolitic imidazolate framework). 13 2 The stepped shape 3 isotherm was rationally attributed to the gas-pressure-responsive behaviour of the ZIF-7 particles, 4 however, only very few data points have been recorded in the step region of the isotherm and the origin 5 of the step was not investigated experimentally. Band the origin 5 of the step was not investigated experimentally”*

This statement is disingenuous since the origin of the step behaviour has previously been well characterised in the solid state. The authors must reword appropriately.

Response

The section of the sentence “*and the origin of the step was not investigated experimentally*” has now been removed from the manuscript.

- *Moreover, the previous study by Lai et al. already suggested that at least ZIF-7 is a suitable material for the development of breathing PLs.13”*

Again this is disingenuous because the Lai work clearly showed that step-changes can be incorporated into PLs, it was not just “suggested”. The authors must reword appropriately.

Response

The sentence has now been reworded as follows: Moreover, the previous study by Lai et al. already demonstrated that at least ZIF-7 is a suitable material for the development of breathing PLs

Following these changes I will recommend publication.

Reviewer #2 (Remarks to the Author):

The authors did an excellent job of addressing my previous comments. Though the novelty of this work is tempered by a previous study of ZIF-7 dispersions, I think there is sufficient novelty here to warrant publication in Nature Communications.

Response

We are particularly pleased that the reviewer is satisfied with our revisions. We truly believe that the reviewers comments/suggestions improved significantly our work and we would like to thank them.

Reviewer #4 (Remarks to the Author):

Following on from the most recent round of peer review, this reviewer believes that the authors have addressed all of the reviewer comments, the science is sound and thorough, including additional experimental evidence.

Response

We tried to address all of the reviewer comments thoroughly and we are happy that the reviewer recognized our effort. We would like to thank the reviewer for the valuable feedback in the previous review rounds and its positive effect on the quality of our work.